# From Pixels to Histopathology: A Graph-Based Framework for Interpretable Whole Slide Image Analysis

**Alexander Weers**[1,2]                     ALEXANDER.WEERS@TUM.DE
**Alexander H. Berger**[1,3]
**Laurin Lux**[1,2,3]
**Peter Schüffler**[1,2,4,5]
**Daniel Rueckert**[1,2,5,6]
**Johannes C. Paetzold**[3,7]             JPAETZOLD@MED.CORNELL.EDU

[1] *TUM School of Computation, Information and Technology, Technical University of Munich, Munich, DE*

[2] *Munich Center for Machine Learning (MCML), Munich, DE*

[3] *Department of Radiology, Weill Cornell Medicine, Cornell University, New York City, NY, US*

[4] *Institute of Pathology, TUM School of Medicine and Health, Technical University of Munich, Munich, DE*

[5] *Munich Data Science Institute (MDSI), Technical University of Munich, Munich, DE*

[6] *Department of Computing, Imperial College London, London, UK*

[7] *Cornell Tech, New York, NY, US*

**Editors:** Accepted for publication at MIDL 2026

## Abstract

The histopathological analysis of whole-slide images (WSIs) is fundamental to cancer diagnosis but is a time-consuming and expert-driven process. While deep learning methods show promising results, dominant patch-based methods artificially fragment tissue, ignore biological boundaries, and produce black-box predictions. We overcome these limitations with a novel framework that transforms gigapixel WSIs into tissue-boundary aligned graph representations and is interpretable by design.

Our approach builds graph nodes from tissue regions that respect natural structures, not arbitrary grids. We introduce an adaptive graph coarsening technique, guided by learned embeddings, to efficiently merge homogeneous regions while preserving diagnostically critical details in heterogeneous areas. Each node is enriched with a compact, interpretable feature set capturing clinically-motivated priors. A graph attention network then performs diagnosis on this compact representation.

We demonstrate strong performance on cancer staging and survival prediction, outperforming methods with similar data requirements. Crucially, our data-efficient model (requiring $> 300\times$ less training data) achieves results competitive with a massive foundation model, while offering full interpretability through feature attribution. Our code is publicly available at https://github.com/aweers/pix2pathology.

**Keywords:** Whole Slide Image, Computational Pathology, Interpretable AI, Graph Representation Learning

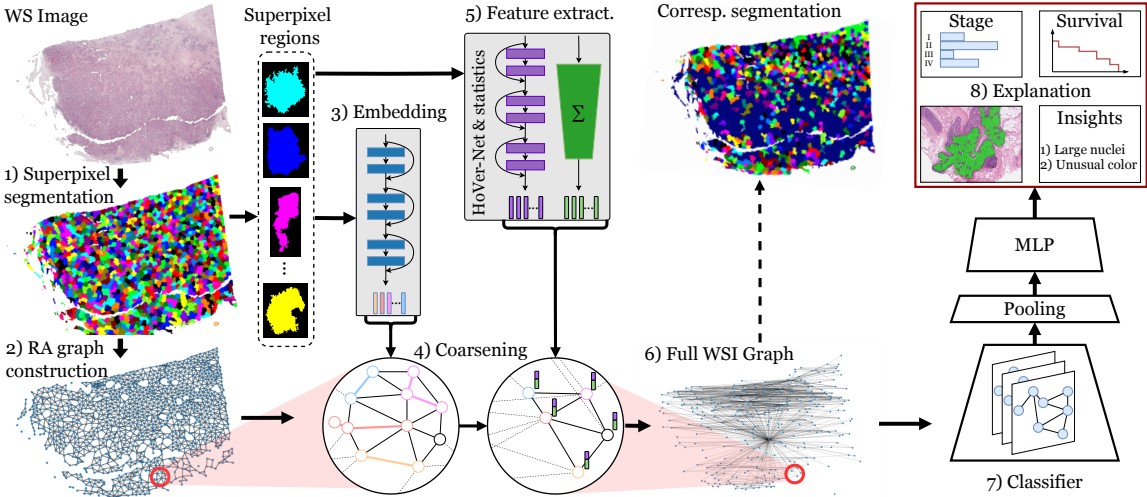

Figure 1: Overview of the pipeline. (1-2) Superpixels are segmented and converted to an RA graph. (3-4) Nodes are merged based on embedding similarity, visualized as a merged version of the initial superpixel segmentation (corresp. segmentation). (5-6) Nodes are enriched with interpretable features (texture, morphology, nuclear). (7) GATv2 performs classification. (8) Explainability with respect to regions and features.

## 1. Introduction

Histopathological analysis of tissue samples is the cornerstone of cancer diagnosis and treatment but is increasingly strained by rising cancer incidence and a limited number of specialized pathologists (Metter et al., 2019). This growing bottleneck motivates the pressing need for reliable computational methods to assist clinical workflows by screening slides, quantifying morphology, and highlighting diagnostically relevant regions.

Whole Slide Images (WSIs) capture tissue at a gigapixel scale (often exceeding 100,000 × 100,000 pixels) and contain multi-scale morphological cues that are crucial for diagnosis (Tseng et al., 2023). Modern deep learning pipelines typically handle that scale by partitioning WSIs into small, fixed-size rectangular patches and processing those patches independently (Van der Laak et al., 2021; Song et al., 2023; Raciti et al., 2023). While practical, this patch-based paradigm has two major limitations. First, rigid patches artificially fragment biological structures (e.g., glands, tumor fronts), destroying natural boundaries and the contextual relationships pathologists use (Ciga et al., 2021). Second, many patch-aggregation methods produce predictions that are hard to attribute to clearly delineated regions or interpretable features, which reduces trust and slows clinical adoption.

We address these limitations by introducing a novel framework that transforms WSIs into morphology-aware graphs that are interpretable by design. In this paper, we propose a graph-based framework that converts WSIs into biologically meaningful graphs, where nodes correspond to tissue regions aligned with natural boundaries, edges capture spatial

contiguity, and node descriptors comprise a compact, clinically interpretable set of texture, morphological, and nuclear features. Our contributions are:

1. A novel pipeline (see Figure 1) that builds graphs from adaptive tissue regions, preserves structural context, and enriches regions with a compact and interpretable feature vector, enabling clinically relevant explanations.

2. An embedding-guided graph coarsening strategy that aggregates homogeneous areas for computational efficiency while retaining high granularity in complex regions.

3. A demonstration that our compact, interpretable model outperforms comparable methods and achieves performance competitive with a large foundation model, while requiring a fraction of the data (300x less) and computational resources.

## 2. Related work

**Patch-based Multi Instance Learning (MIL)**. The dominant paradigm in WSI analysis treats slides as bags of fixed-sized patches (Srinidhi et al., 2021; Shmatko et al., 2022). While seminal models like DeepSets (Zaheer et al., 2017), DSMIL (Li et al., 2021), and the attention-based ABMIL (Ilse et al., 2018) established the effectiveness of this approach, they discard the spatial topology of the tissue.

Recent foundation models, such as UNI2-h (Chen et al., 2024) and CHIEF (Wang et al., 2024a) scale this approach by pre-training on massive datasets. However, these methods usually rely on learned latent embeddings that are not interpretable (Kaczmarzyk et al., 2024). While recent efforts like Si-MIL (Kapse et al., 2024) have introduced feature interpretability, they, along with standard MIL, remain constrained by the rigid patch grid, which artificially fragments biological structures such as tumor boundaries.

**Graph-based WSI Analysis**. To restore spatial context, Graph Neural Networks (GNNs) have been adopted to model the tissue microenvironment, having already demonstrated advantages across various other medical domains (Wu et al., 2021; Qiu et al., 2024; Lux et al., 2025; Li et al., 2025). Approaches like Patch-GCN (Chen et al., 2021) and GraphTransformer (Zheng et al., 2022) construct graphs where nodes are patches and edges represent adjacency. Extensions like DM-GNN (Wang et al., 2024b) further refine this by modeling morphological and global dependencies to capture complex patch correlations. While this improves context modeling, the nodes remain defined by an arbitrary grid, compromising the biological fidelity of the graph structure.

**Biologically-Aligned Representations**. Moving beyond grids, Cell-Graphs (Zhou et al., 2019; Pati et al., 2022) model individual cells as nodes. While highly granular, this approach faces severe scalability issues, often generating graphs with millions of nodes that are computationally prohibitive for whole-slide analysis. Superpixel methods offer a middle ground by clustering pixels into perceptually meaningful regions (Zormpas-Petridis et al., 2021; Luo et al., 2025), but typically still rely on latent features that are not interpretable.

**Our Contribution**. We bridge these gaps by proposing a structure-adaptive graph framework that respects natural tissue boundaries. By employing adaptive coarsening, we efficiently aggregate homogeneous tissue while preserving granularity in heterogeneous regions. Crucially, we utilize interpretable features by design rather than opaque embeddings, ensuring clinical transparency.

## 3. Method

Our framework transforms WSIs into compact, interpretable graphs, suitable for clinical tasks and explainable predictions. This transformation addresses the core challenges of computational pathology: the immense scale of the data, the need for both local features and global connections, and the benefit of explainable models in a clinical context.

The proposed pipeline( see Figure 1) constitutes a multi-stage process that moves from a low-level pixel representation to a high-level graph suitable for clinical prediction tasks.

The pipeline consists of four main steps. First, the WSI is segmented into a large set of small, biologically aligned regions using superpixels. Second, these fine-grained regions are adaptively merged based on their semantic similarity, creating a coarsened graph that reflects the macroscopic tissue organization. Third, each node in this coarsened graph is enriched with a comprehensive set of interpretable, domain-informed features that capture texture, morphology, and nuclear characteristics. Finally, a Graph Attention Network (GAT) is applied to this final graph representation to perform slide-level classification tasks.

### 3.1. Superpixel segmentation

The first step of our framework is to convert the raw pixel data of a WSI into a meaningful set of initial regions that respect the boundaries of tissue structures. More formally, given a WSI $\mathcal{I} \in \mathbb{R}^{H \times W \times 3}$, we identify tissue foreground $\mathcal{T} \subset \mathcal{I}$ by applying Otsu's thresholding on the HSV saturation channel and morphological cleaning, as described in (Lu et al., 2021b).

With the tissue area $\mathcal{T}$ identified, we then partition it into an initial set of small, perceptually meaningful regions. Instead of using a rigid grid of square patches, we segment the tissue area into superpixels, $\mathcal{S} = \{s_1, ..., s_K\}$, using Simple Linear Iterative Clustering (SLIC) (Achanta et al., 2012) on a low-resolution version of the WSI ($0.625\times$ magnification). SLIC is a variant of k-means clustering, grouping spatially close pixels with similar colors, creating small, irregularly shaped regions that naturally adhere to local tissue boundaries (see Figure 3), such as gland edges as the interface between tumor and stroma. The number of superpixels $K$ is chosen to target an average size of $300 \times 300$ pixels at a magnification level of x32. This oversegmentation forms the basis of our initial region-adjacency (RA) graph $\mathcal{G}_0 = (\mathcal{V}_0, \mathcal{E}_0)$, where each node $v_i \in \mathcal{V}_0$ corresponds to a superpixel $s_i$, and an edge $(v_i, v_j) \in \mathcal{E}_0$ exists if superpixels $s_i$ and $s_j$ are spatially adjacent.

### 3.2. Adaptive graph coarsening

The initial graph $\mathcal{G}_0$ is too fine-grained for efficient downstream processing. To address this, we introduce an adaptive graph coarsening procedure designed to merge semantically similar, adjacent regions into larger super-nodes. The goal is to produce a more abstract, computationally efficient graph, where nodes can represent larger, homogeneous tissue components. The *adaptive* nature of this process is key: it preserves high granularity in heterogeneous, complex regions while substantially simplifying large, uniform areas, thus reducing graph size while preserving the most important morphological boundaries.

**Region embeddings** The foundation of our coarsening strategy is a semantically rich representation for each graph node. We generate a region embedding $h_i \in \mathbb{R}^{512}$ for each node $v_i$ using a ResNet-18 feature encoder. The feature encoder was pretrained on his-

tology patches using contrastive learning, enabling it to capture fine-grained textural and morphological patterns (Zheng et al., 2022). To adapt the irregularly shaped superpixels to the rectangular input required by the ResNet, we extract the bounding box of the region from the WSI and zero-mask (black out) all pixels outside the superpixel boundary. This ensures the input matches the ResNet's rectangular requirements while isolating the visual features of the specific region.

**Greedy merging** With these embeddings, we perform an agglomerative merging of nodes. We compute the cosine similarity $S_{ij}$ for adjacent node pairs $(v_i, v_j)$ in the initial graph $\mathcal{G}_0$.

The merge process is performed greedily, starting with the pair of the highest similarity. Nodes are merged sequentially until the score of the next candidate pair falls below the predefined threshold $\tau$. The similarity threshold $\tau$ is a hyperparameter that controls the final granularity of the graph.

When two nodes are merged, they are replaced by a single new node whose region is the union of the original nodes, and which inherits the edges of its predecessors. Crucially, we do not recompute or average the embeddings for the newly formed region during the process. The decision to merge two adjacent regions is determined solely by the similarity of their initial, local embedding. This allows the algorithm to aggregate larger homogeneous areas that may exhibit gradual feature shifts (e.g., due to staining gradients) by relying on the local similarity of adjacent components. This agglomerative process continues until no adjacent nodes have a similarity exceeding $\tau$. The result is a coarsened graph $\mathcal{G} = (\mathcal{V}, \mathcal{E})$ where $|\mathcal{V}| \leq |\mathcal{V}_0|$.

This adaptive coarsening ensures that our final graph is a compact yet faithful representation of the tissue's macro-architecture.

### 3.3. Interpretable node features

While the learned region embeddings $h_i$ used for coarsening are powerful, we do not use them as node features since they are inherently black-box and lack direct clinical interpretability. To create a framework that is explainable by design, we engineer a separate set of domain-informed features that describe each node in the final coarsened graph.

The feature vector $x_i = [x_i^{\text{nuc}\top}, x_i^{\text{tex}\top}, x_i^{\text{morph}\top}]^\top \in \mathbb{R}^{191}$ is a concatenation of three distinct feature groups, capturing nuclear, texture and intensity, and morphology and color features:

- **Nuclear features** $x_i^{\text{nuc}} \in \mathbb{R}^{77}$: Nuclear morphology is a cornerstone of histopathological assessment. We leverage a pretrained HoVerNet model (Graham et al., 2019), a frequently-used deep learning model for simultaneous nuclear instance segmentation and classification. For each region, we apply HoVerNet to detect individual nuclei and classify them into one of six types (see Appendix D). We then compute a rich set of statistics, including count, density, size, and shape characteristics for each nucleus type, providing a quantitative summary of the cellular composition of the region.

- **Texture and intensity features** $x_i^{\text{tex}} \in \mathbb{R}^{93}$: This group quantifies the micro-patterns within the tissue region. It includes features derived from the Gray-Level Co-occurrence Matrix (GLCM), such as contrast, correlation, and energy, which capture the spatial relationships between pixel intensities. It also includes Local Binary Pat-

tern (LBP) features and general intensity as defined in the Pyradiomics library (van Griethuysen et al., 2017).

- **Morphological and color features** $x_i^{\mathrm{morph}} \in \mathbb{R}^{21}$: This set of features describes the size of the region and its color properties. We compute color distribution statistics (mean, variance) across multiple color spaces (RGB, HSV, LAB). These features capture tissue morphology.

To reduce redundancy in the high-dimensional initial feature vector, we perform correlation-based pruning. We iteratively remove one feature from any pair in the training data whose absolute Pearson correlation $|\rho|$ exceeds a threshold $\xi$, resulting in a more compact and robust feature set. This feature pruning is done once for the whole dataset and not for each individual sample. A complete list of all features can be found in Appendix J.

The set of node features $\mathcal{X} = \{x_i \mid v_i \in \mathcal{V}\}$ extends the previously defined coarsened graph to form the final, compact graph representation $\mathcal{G} = (\mathcal{V}, \mathcal{E}, \mathcal{X})$ of the WSI $\mathcal{I}$.

### 3.4. Graph attention network for classification

We employ a Graph Attention Network v2 (GATv2) (Brody et al., 2022), which takes our compact representation $\mathcal{G}$ as input and learns the relative importance of neighboring nodes for the final prediction.

After passing a node through a stack of GAT layers, we obtain a set of high-level node embeddings that are context-aware, incorporating information from their local neighboring tissue environment. To arrive at a single slide-level prediction, a final graph readout function is applied. This function aggregates all the node embeddings using a mean-pooling into a single graph-level feature vector, which is then fed into a standard multilayer perceptron (MLP).

### 3.5. Explainability

To provide transparent explanations for the predictions, we employ Integrated Gradients (IG) (Sundararajan et al., 2017). IG attributes the prediction output to input features by integrating gradients along a linear path from a baseline (zero vector) to the actual input. This yields a mathematically unique attribution score matrix of the same shape as the input node features $\mathcal{X}$. We process these scores in two ways:

1. To identify influential regions, we sum the absolute attribution scores across the feature dimension for each node, highlighting superpixels with the highest aggregate contribution.

2. To interpret biological drivers, we rank specific features (e.g., nuclear density, skewness) based on the magnitude of their attribution scores within the most important regions.

## 4. Results & Discussion

We evaluate our method on two challenging clinical tasks: cancer stage classification and patient survival prediction. Our experimental framework, including the datasets, task definitions, and baseline models used for comparison, is detailed below.

| | Method | Stage Classification | | | Survival Prediction |
|---|---|---|---|---|---|
| | | AUC ↑ | F1$_m$ ↑ | Bal. Acc ↑ | C-Index ↑ |
| **TCGA-BRCA** | *Comparable Baselines* | | | | |
| | DeepSets | 52.3*(3.05) | 15.8* (5.50) | 25.4 (0.10) | 48.9*(2.47) |
| | ABMIL | 61.8*(5.64) | 20.2* (2.13) | 26.0 (1.11) | 61.7 (1.81) |
| | GraphTransformer | 63.3 (2.33) | 18.6* (0.89) | 24.9 (0.14) | 52.7*(5.61) |
| | DM-GNN | 62.5*(2.19) | 23.2* (5.40) | **28.6** (4.34) | 59.9 (5.22) |
| | Ours | **67.2** (3.08) | **28.0** (8.24) | 27.0 (2.54) | **62.9** (3.67) |
| | *Foundation model* | | | | |
| | UNI2-h | 69.8 (3.05) | 30.1 (4.45) | 31.0*(2.78) | 62.1 (6.90) |
| **TCGA-UCEC** | *Comparable Baselines* | | | | |
| | DeepSets | 49.4*(1.48) | 9.46*(6.47) | 24.4 (1.67) | 50.3*(0.60) |
| | ABMIL | 49.0*(2.61) | 18.8 (0.00) | 25.0 (0.00) | 58.3 (5.31) |
| | GraphTransformer | 54.1 (4.51) | 18.7 (1.18) | 25.2 (0.58) | 57.5 (7.84) |
| | DM-GNN | 55.3 (3.82) | **20.6** (2.61) | 26.3 (3.19) | 55.9 (8.38) |
| | Ours | **56.4** (4.17) | 19.5 (1.19) | **26.5** (2.37) | **60.0** (0.97) |
| | *Foundation Model* | | | | |
| | UNI2-h | 62.7*(5.03) | 28.0*(4.67) | 30.4*(2.92) | 62.8 (5.38) |

Table 1: Performance on TCGA-BRCA and TCGA-UCEC. Best performance among comparable baselines is shown in **bold**, second best underlined. UNI2-h is listed separately as it was pretrained on over 350,000 external WSIs. Asterisks (*) denote statistically significant differences ($p < 0.05$) to our method.

## 4.1. Experimental Setup

**Datasets and Tasks.** We evaluate our method on two TCGA cohorts: Breast Invasive Carcinoma (BRCA, $N = 1,101$) (Lingle et al., 2016) and Uterine Corpus Endometrial Carcinoma (UCEC, $N = 539$) (Erickson et al., 2016). We benchmark performance on two clinical tasks: (1) Tumor Stage Classification, determining pathological stage directly from WSIs, and (2) Patient Survival Prediction, identifying risk groups for prognostic assessment.

The first task, cancer stage classification, involves predicting the pathological stage of the tumor directly from the WSI. Cancer staging is a critical component of clinical oncology, as it describes the extent of cancer progression and acts as a basis to determine treatment plans. The stage is typically described on a scale from I to IV, with higher stages indicating more advanced disease. Since the task is a multi-class problem, we use cross-entropy loss for all models.

The second task, survival prediction, stratifies patients into risk groups based on their predicted prognosis. Accurate survival prediction is essential for personalizing treatment, managing patient care, and identifying high-risk individuals who may benefit from more aggressive therapies. For training we use Negative log-likelihood survival loss (Zadeh and Schmid, 2020). We include the full details on all hyperparameters in Appendix F.

**Baselines.** We compare our framework against two categories of methods: The first group consists of efficient baselines and includes representative MIL methods (DeepSets (Zaheer et al., 2017), ABMIL (Ilse et al., 2018)) and graph-based approaches (GraphTransformer (Zheng et al., 2022), DM-GNN (Wang et al., 2024b)) that utilize a comparable amount of data and have a similar model size. To represent standard deep learning based histopathology workflows, we utilize established pre-trained encoders for these baselines: GraphTransformer employs its original patch encoder, while DeepSets, ABMIL, and DM-GNN utilize the widely adopted ResNet-50 features from CLAM (Lu et al., 2021a), following the protocol in (Wang et al., 2024b). The second comparison category features UNI2-h (Chen et al., 2024), a large-scale vision foundation model pre-trained on more than 100 million pathology patches. By leveraging massive data and computational resources to learn highly generalizable morphological representations, UNI2-h provides an upper-bound benchmark that contextualizes the performance of our method.

**Evaluation Protocol.** For all methods we run five independent hyperparameter sweeps on different training data splits, select a model based on validation set performance and evaluating on a separate test set (see Appendix C for more details). All results report the mean and standard deviation across five independent folds with site-level splitting to minimize batch effects (Howard et al., 2021).

For staging, we report Area Under the Receiver Operating Characteristic Curve (AUC), Balanced Accuracy, and macro-averaged F1-score ($F1_m$) to account for class imbalance. For the survival prediction, we report the Concordance Index (c-index) (Harrell et al., 1982).

### 4.2. Performance Analysis

Table 1 summarizes the performance of our proposed framework across cohorts and tasks. Our method establishes a new state-of-the-art among resource-efficient baselines, outperforming existing MIL and graph-based approaches on the majority of primary metrics across both datasets.

**Comparison with Efficient Baselines.** In the Stage classification task on TCGA-BRCA, our method achieves the highest AUC and $F1_m$ scores. We report an AUC of 67.2, surpassing the strongest baseline (GraphTransformer) by a clear margin of 3.9 points. Notably, our $F1_m$-score of 28.0 represents a relative improvement of over 20% compared to the runner-up (DM-GNN: 23.2). This dominance extends to survival prediction, where we outperform all comparable baselines (c-index: 62.9 vs 61.7).

The trend of strong performance continues on the smaller TCGA-UCEC dataset, demonstrating the robustness and good generalization of our approach. It achieves the highest AUC (56.4) and leads in survival prediction (c-index: 60.0 vs 58.3). While the margins are tighter on this smaller cohort, our method shows consistently strong performance and is only close second in the $F1_m$-score (19.5 vs 20.6 DM-GNN).

We attribute this advantage to our fundamentally different approach to represent tissue. Standard MIL models discard spatial context, while patch-graph methods often impose artificial grids that fragment biological entities. By aligning nodes with natural tissue boundaries and enriching them with interpretable clinical features, our model captures subtle, stage-determining morphological signals that grid-constrained topologies miss.

| | Method Configuration | | Stage Classification | | | Survival |
|---|---|---|---|---|---|---|
| | **Model** | **Input Features** | AUC ↑ | F1$_m$ ↑ | Bal. Acc ↑ | C-Index ↑ |
| **BRCA** | ABMIL$^\dagger$ | ResNet-50 | 61.8*(5.64) | 20.2*(2.13) | 26.0 (1.11) | 61.7 (1.81) |
| | ABMIL | Interpretable | 58.9*(5.28) | 16.8*(4.49) | 26.7 (2.95) | 51.0*(2.30) |
| | Graph | ResNet-50 | 57.9*(5.65) | 19.1*(0.37) | 25.1 (0.12) | 51.4*(1.14) |
| | **Graph$^\dagger$** | **Interpretable (Ours)** | **67.2** (3.08) | **28.0** (8.24) | **27.0** (2.54) | **62.9** (3.67) |
| **UCEC** | ABMIL$^\dagger$ | ResNet-50 | 49.0*(2.61) | 18.8 (0.00) | 25.0 (0.00) | 58.3 (5.31) |
| | ABMIL | Interpretable | 52.1 (2.74) | 17.6 (3.59) | 25.1 (1.63) | 51.3 (2.87) |
| | Graph | ResNet-50 | 52.0 (3.49) | **20.2** (1.73) | 25.4 (0.92) | 59.6 (4.18) |
| | **Graph$^\dagger$** | **Interpretable (Ours)** | **56.4** (4.17) | 19.5 (1.19) | **26.5** (2.37) | **60.0** (0.97) |

Table 2: Ablation study on TCGA-BRCA and TCGA-UCEC isolating the contribution of the Graph-based structure and the Interpretable features. $\dagger$ denotes rows that are copied from Table 1 for easier comparison.

### 4.3. Efficiency vs. Foundation Models

While foundation models rely on scale, our approach demonstrates that an intelligent representation design can achieve competitive performance with a fraction of the resources. On TCGA-BRCA, our model marginally exceeds the performance of UNI2-h in survival prediction (c-index: 62.9 vs 62.1) and remains statistically comparable in staging. This parity is achieved despite massive disparities in resource usage (see Appendix E for details):

- **Data scale**: Including the training of models used for feature extraction, our method still consumed $300\times$ less data compared to UNI2-h.

- **Feature interpretability**: Beyond efficiency, our model is interpretable by design. Predictions can be traced back to specific, biologically-grounded regions and a curated set of clinically-motivated features. This stands in stark contrast to the black-box features used by most other models and UNI2-h.

### 4.4. Ablation Studies

To validate the contribution of each component in our proposed pipeline, we conducted a series of ablation studies on the TCGA-BRCA dataset. We investigate four critical design dimensions: impact of architecture vs. features, feature redundancy pruning ($\xi$), graph topology construction via region merging ($\tau$), and the synergy between different feature modalities. The results for the architecture vs feature ablation are shown in Table 2, while the other results are summarized in Table 3.

**Architecture vs features:** To disentangle the performance gains attributed to our graph-based method from those driven by the interpretable feature set, we evaluated all combinations of architectures and features (see Table 2). We compared our proposed framework (Graph + Interpretable) against the standard MIL baseline (ABMIL) and the widely used,

| | Setting / Features | AUC ↑ | F1$_m$ ↑ | Bal. Acc ↑ |
|---|---|---|---|---|
| $\xi$ | 0.95 *(aggressive pruning)* | 62.9 (2.02) | 18.6 (0.25) | 28.1 (3.22) |
| | 0.99 | **65.4** (2.96) | **25.4** (2.99) | **28.6** (1.88) |
| | 1.00 *(no pruning)* | 61.8 (5.43) | 19.9 (1.30) | 25.5 (0.53) |
| $\tau$ | 0.5 *(coarse regions)* | 59.5 (2.29) | 18.6 (0.26) | 25.0 (0.00) |
| | 0.8 | **65.0** (4.65) | 20.3 (2.81) | 26.6 (1.96) |
| | 0.9 | 64.7 (5.55) | 18.6 (0.26) | 25.0 (0.00) |
| | 0.95 | 64.3 (5.10) | **25.1** (4.15) | **28.4** (2.46) |
| | 1.0 *(highest granularity)* | 60.4 (4.98) | 22.1 (2.64) | 25.8 (0.53) |
| **Features** | Nuc, Morph, Tex | **65.4** (3.78) | **25.3** (3.17) | **28.4** (2.46) |
| | Nuc, Morph | 65.3 (3.74) | 19.8 (1.30) | 25.2 (0.60) |
| | Nuc,         Tex | 64.7 (2.49) | 21.2 (1.78) | 26.2 (1.32) |
| | Nuc | 62.9 (5.30) | 23.7 (2.09) | 27.6 (1.93) |
| | Morph | 61.8 (1.77) | 21.1 (1.71) | 26.7 (0.97) |
| | Morph, Tex | 57.3 (3.52) | 20.0 (1.01) | 25.1 (0.59) |
| | Tex | 55.7 (2.78) | 18.8 (0.56) | 25.0 (0.45) |

Table 3: Combined ablation studies for TCGA-BRCA. Sections ($\xi$: correlation threshold, $\tau$: group similarity, Features: feature combinations) report mean validation AUC, F1$_m$, and balanced accuracy. Bold marks the best in each category.

learned patch embeddings from a pre-trained ResNet-50 (CLAM (Lu et al., 2021b)). Three key insights emerge from this analysis.

First, we observe that our method relies on both the graph architecture and the feature set, showing the strongest performance in this configuration. When using interpretable features within a standard ABMIL framework, performance drops significantly on both BRCA (AUC: 67.2 to 58.9; C-Index: 62.9 to 51.0) and UCEC (AUC: 56.4 to 52.1; C-Index: 60.0 to 51.3). This indicates that local, interpretable statistics (such as nuclear density or texture) lose their predictive value for these challenging tasks when aggregated globally without spatial context.

Second, the interpretable feature set can be a replacement for learned embeddings in a MIL setting when applied to the task of stage prediction. The performance is comparable, but the interpretability of such a combination would be greatly improved. However, we observe that this is not the case of survival prediction, where interpretable features in a MIL setting perform significantly worse compared to ResNet-50 embeddings on both tasks.

Third, using learned features in combination with a graph instead of a MIL architecture, does neither hurt nor boost performance significantly, with the exception of survival prediction on BRCA where MIL clearly performs better (ABMIL: 61.7 vs Graph: 51.4).

**Graph Construction ($\tau$):** The region merging threshold $\tau$ exhibits a trade-off between abstraction and detail. Lower values of $\tau$ merge larger, more heterogeneous regions, while higher values preserve more fine-grained details. We observe that merging regions too aggressively ($\tau = 0.5$) leads to significant information loss. Additionally, preserving every superpixel ($\tau = 1.0$) creates an overly dense graph, increasing computational complexity and introducing high-frequency noise. A moderate $\tau = 0.95$ balances the simplification

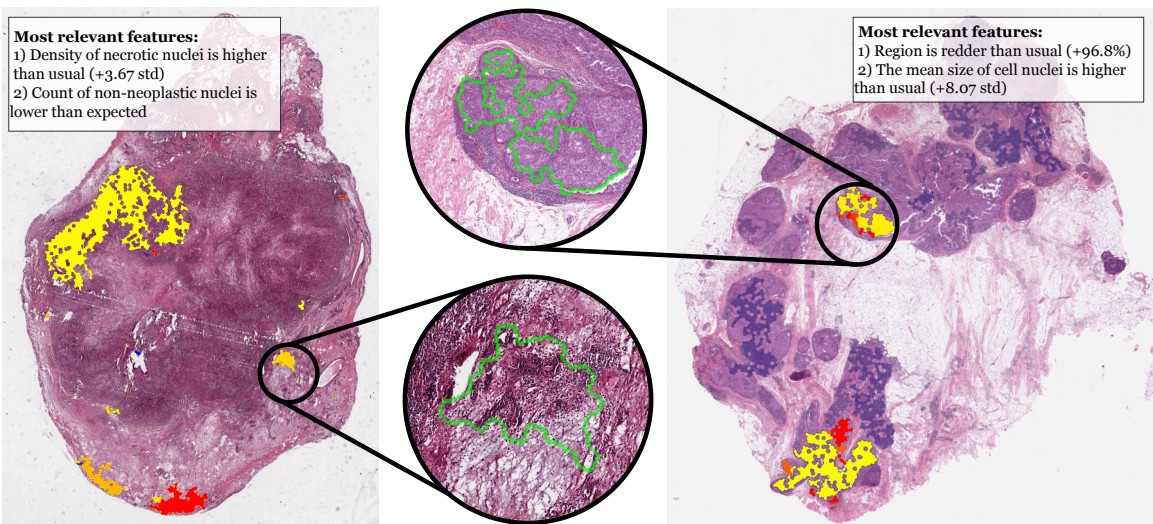

Figure 2: Explanations for two TCGA-BRCA Stage 2 samples. Red overlays indicate influential regions (via Integrated Gradients). We also list the top-attributed interpretable features compared to dataset statistics to highlight biological drivers.

of homogeneous areas with the preservation of heterogeneous details, yielding a strong performance across all metrics.

**Feature Pruning ($\xi$):** The correlation threshold $\xi$ governs the trade-off between feature dimensionality and information retention. A high correlation threshold ($\xi = 0.99$) effectively removes redundant features without losing information, whereas retaining all features ($\xi = 1.0$) leads to slight overfitting. Conversely, setting the threshold too low ($\xi = 0.95$) overly simplifies the feature space, discarding subtle but discriminative signals.

**Feature Groups:** The full combination of all three modalities yields the best overall performance (65.4 AUC, 25.3 F1$_m$), confirming the hypothesis that a holistic view of the tissue microenvironment is superior to any single descriptor. While nuclear features are the most discriminative individual group (62.9 AUC), the inclusion of texture and morphology significantly boosts the overall performance and robustness to class imbalances, confirming the value of a holistic tissue representation.

Since the cost of multiple runs with different similarity thresholds is comparatively low, we would recommend it when the framework is applied to new datasets.

## 4.5. Qualitative Analysis and Interpretability

The goal of computational pathology is not just predictions, but actionable insights. A key motivation for moving beyond patch-based black boxes is the need for trustworthy and clinically relevant explanations. Our framework delivers on this promise.

Leveraging the attribution scores derived via Integrated Gradients (see Section 3.5), we attribute predictions to specific tissue regions and feature sets (see Figure 2). We note that Integrated Gradients, being an additive attribution method, can not highlight interactions

of distinct regions or features. However, since the method models interactions of regions and features (via graph layer), the final attribution score represents a node's contribution given the graph structure and the feature interactions learned by the model.

To validate the clinical utility of these explanations, we conducted a preliminary qualitative review with a pathology resident. The expert confirmed that the regions identified by the model as most important align closely with diagnostically relevant tissue structures. Specifically, the model frequently highlighted direct tumor tissue and inflammatory infiltrates at tumor boundaries, both of which are critical for staging and prognosis. Furthermore, the expert noted that the model's tendency to focus on a limited number of small, informative regions mirrors actual clinical workflows, where diagnosis is often driven by distinct morphological regions rather than a uniform assessment of the whole slide.

The attribution to specific interpretable features also received positive validation. The expert noted that explicitly using nuclear statistics, such as the count and size deviations of specific subtypes (e.g., neoplastic vs. inflammatory), provides highly informative, clinically grounded evidence. For instance, in Figure 2 (left), identifying regions with high density of necrotic nuclei as Stage 2 drivers aligns with common grading criteria.

However, this expert review also underscored the necessity of critical oversight. In Figure 2 (right), while the model correctly flagged "unusual color" (redness) as a statistical deviation contributing to the prediction, the pathologist cautioned that such color variations can sometimes stem from staining artifacts rather than biological signals. This distinction highlights the value of a transparent feature set: by explicitly naming "color" as the driver, the model allows the expert to accept valid morphological signals (like nuclear size) while potentially discounting artifacts, a level of scrutiny impossible with black-box embeddings.

As shown in these examples, we further contextualize these explanations by comparing the most attributed features to statistics derived from the training dataset. This allows experts to immediately discern whether a prediction is driven by a deviation from common patterns or the presence of a specific, rare indicator, thereby building confidence in the model's findings. We provide additional qualitative examples in Appendix B.

## 5. Conclusion

We presented a WSI analysis framework that bridges high-performance deep learning with clinical interpretability. By shifting from rigid patches to adaptive, biologically-grounded graphs, we preserve essential tissue structure while reducing computational cost. Enriching nodes with domain-specific nuclear and texture features allows predictions to be traced back to valid morphological evidence, aligning AI explanations with diagnostic workflows.

Empirically, our method outperforms comparable baselines and rivals the massive UNI2-h foundation model, demonstrating that resource-efficient, interpretable designs can compete with large-scale pre-training.

**Limitations and future work.** Future iterations will address the computational overhead of the nuclei detection preprocessing. We also plan to validate the clinical utility of our explanations through expert reader studies, explore extensions to other staining variations, and leverage the graph structure to integrate multi-modal genomic data.

## Acknowledgments

We thank Dr. med. Franz-Leonard Klaus, for serving as a medical expert in the qualitative evaluation and providing helpful insights into clinical workflows. This work was partially supported by the German Federal Ministry of Research, Technology and Space (BMFTR) as part of the Software Campus 3.0 (TU München) under grant number 01IS23069.

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

## Appendix A. Tissue-adaptive segmentation with superpixels

To show that superpixels created with SLIC (Achanta et al., 2012) respect tissue boundaries, we show an overlay of such a segmentation on the tissue in Figure 3. The individual regions are irregularly formed, and their borders closely align with natural tissue structures, making it well-suited for our use case of serving as initial node creation.

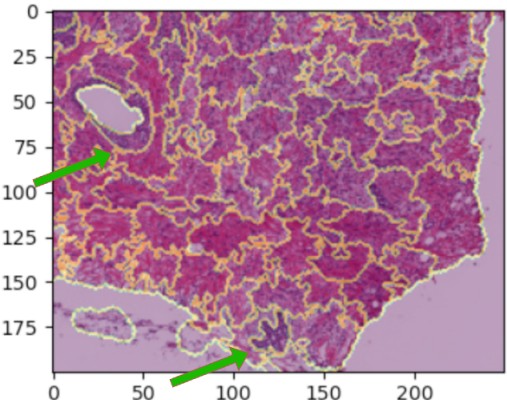

Figure 3: Illustration of fine-grained, tissue-adaptive segmentation. Superpixel boundaries, highlighted in yellow, are overlaid on a tissue micrograph to demonstrate their precise alignment with the underlying morphological structures.

## Appendix B. Additional qualitative examples

We show additional qualitative examples in Figure 4 and Figure 5 with highlighted regions in global slide context, a zoomed-in view and view of important features in such regions. We observe, that just a few critical regions are highlighted as they are the main drivers for the prediction. Additionally, we note that out of the 191 features only a handful are major factors for the prediction, serving as a brief explanation accompanying the prediction.

## Appendix C. Experimental setup

To ensure a rigorous and unbiased evaluation, we first partition the entire dataset into a fixed training set and a hold-out test set. All splits are performed at the location- (and therefore also patient-) level to prevent data leakage. Model optimization and selection are conducted solely on the training set. For each method, we perform a 25-trial random search for the learning rate and weight decay. Each trial consists of training five model instances on a sub-partition of the training set and evaluating them on a validation set. The models from the trial yielding the best average validation performance are then selected and their performance on the hold-out test set is reported (mean and standard deviation). To asses statistical significance between our method and the baselines, we apply an independent Student's t-test to the sets of five test scores.

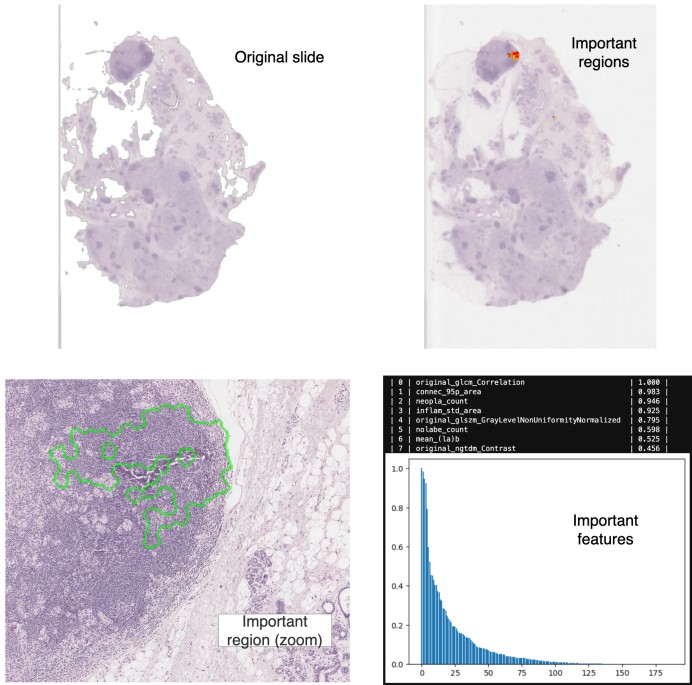

Figure 4: Top left: Original WSI; top right: important regions are highlighted; bottom left: zoomed in version of highlighted region; bottom right: feature attribution scores.

To evaluate models on survival prediction we use the concordance index (c-index) (Harrell et al., 1982). The first task, cancer stage prediction, involves predicting the pathological stage of the tumor directly from the WSI. Cancer staging is a critical component of clinical oncology, as it describes the extent of cancer progression and acts as a basis to determine treatment plans.

The second task, survival prediction, stratifies patients into risk groups based on their predicted prognosis. This task is framed as a multi-class classification task, where patients' risks are discretized into four groups, and this assignment acts as the prediction target, following (Wulczyn et al., 2020). As metric we follow survival prediction literature and calculate the concordance index:

The concordance index (c-index) measures the ratio of correctly ordered (concordant) pairs to the total number of informative pairs. To address significant variations in follow-up duration and median survival across different cancer types, we calculated the c-index for the combined cohort using a stratified aggregation. This involved summing the concordant and informative pairs within each individual study before calculating the final ratio, ensuring that patient rankings were only evaluated relative to others in the same specific study. (Wulczyn et al., 2020)

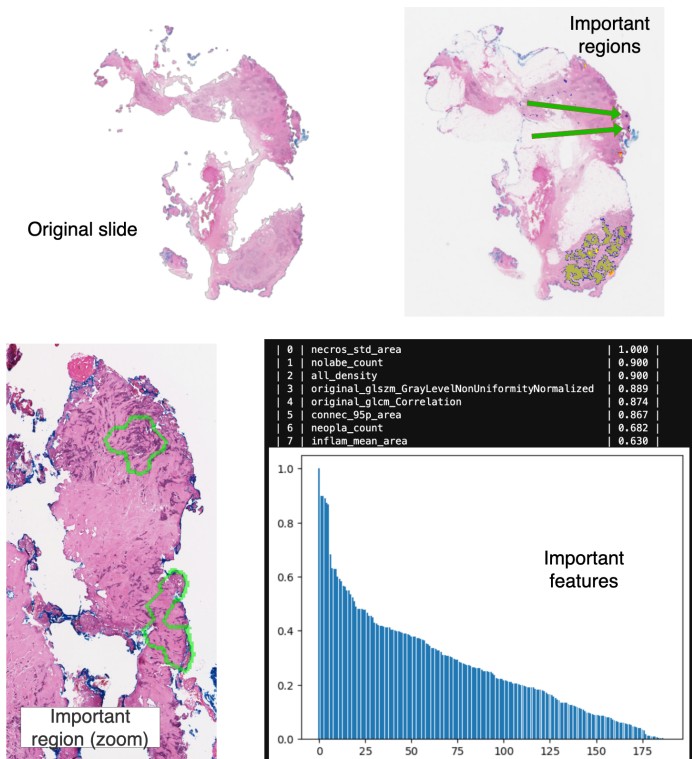

Figure 5: Top left: Original WSI; top right: important regions are highlighted; bottom left: zoomed in version of highlighted region; bottom right: feature attribution scores.

## Appendix D. Nuclear explanations

Hover-Net classifies nuclei into one of six classes that can be used to provide detailed explanations (Gamper et al., 2019):

- Neoplastic: These are the tumor cells themselves (malignant or benign), characterized by abnormal growth.

- Non-Neoplastic Epithelial: These are normal, hyperplastic, or dysplastic epithelial cells that are not part of the tumor mass.

- Inflammatory: Immune system cells, such as lymphocytes and macrophages, responding to the tumor microenvironment.

- Connective / Soft Tissue: Cells forming the stroma and supporting tissue, like fibroblasts and endothelial cells.

- Dead: Nuclei of cells in apoptotic or necrotic states, often fragmented or degraded, which can be an important biological indicator.

## Appendix E. Efficiency calculations

Here are extended calculations to compare the efficiency of our model to UNI2-h with respect to data requirements and model size:

| Model | Data usage (WSIs) | Parameters |
|---|---|---|
| UNI2-h | 350,000$^\dagger$ | 681,394,176 |
| HoVerNet | 455$^\ddagger$ | 37,721,166 |
| Embedding model | 665 | 11,680,936 |
| Preprocessing total | 1,110 | 50,406,010 |

Table 4: Overview of data usage and model parameters in the preprocessing pipelines of UNI-2h and our method.

$^\dagger$ exact number is not reported, model card just mentions "350,000+"
$^\ddagger$ counting each visual field as unique WSI (upper bound)

## Appendix F. Hyperparameter details

Here we list all hyperparameters used for model training.

| Hyperparameter | Value / Range |
|---|---|
| *Fixed Parameters* | |
| Optimizer | Adam ($\beta_1 = 0.9, \beta_2 = 0.999$) |
| Epochs | 20 |
| Activation Function | ReLU |
| MLP Dimension | 384 |
| Graph Layers | 2 |
| Graph Embedding Dimension | 512 |
| *Sweep Configuration* | |
| Learning Rate | Log-uniform $[10^{-4}, 10^{-2}]$ |
| Weight Decay | $\{10^{-3}, 10^{-5}, 10^{-8}\}$ |

Table 5: Hyperparameter settings and search space for model training.

## Appendix G. Statistics of region sizes

In the table below, we show the distribution of region sizes. We observe that there is some variety in height and width (as expected and desired), but all regions stay in a similar range of sizes.

| Metric | Mean | 5% | 25% | Median | 75% | 95% |
|--------|------|-----|-----|--------|-----|-----|
| Height | 188  | 96  | 168 | 192    | 216 | 264 |
| Width  | 194  | 104 | 168 | 192    | 216 | 288 |

Table 6: Distribution of region dimensions ($N = 142,929$). While the dataset contains regions of varying shapes, the interquartile ranges (25%–75%) indicate that the majority of regions fall within a consistent spatial range.

## Appendix H. Additional ablation studies

This section presents additional ablation studies evaluating the impact of the correlation threshold $\xi$ and group similarity $\tau$ on TCGA-UCEC validation performance. Compared with the TCGA-BRCA results, we observe greater robustness across both hyperparameters. Consistent with previous findings, mild feature pruning improves performance; however, unlike in TCGA-BRCA, larger regions (corresponding to lower $\tau$ values) yield better performance.

|       | Setting | AUC ↑ | F1$_m$ ↑ | Bal. Acc ↑ |
|-------|---------|-------|----------|------------|
| $\xi$ | 0.95 *(aggressive pruning)* | **55.5** (3.37) | 17.4 (3.67) | 24.6 (0.78) |
|       | 0.99 | 55.1 (3.64) | **23.9** (5.10) | **28.3** (3.41) |
|       | 1.00 *(no pruning)* | 54.7 (5.17) | 13.9 (0.86) | 26.0 (0.65) |
| $\tau$ | 0.5 *(coarse regions)* | **55.7** (2.29) | 17.2 (0.43) | 25.3 (0.51) |
|       | 0.8 | 53.9 (4.66) | 14.7 (2.71) | **28.3** (3.64) |
|       | 0.9 | 52.4 (3.52) | 19.5 (1.07) | 22.9 (0.94) |
|       | 0.95 | 54.0 (5.15) | 19.0 (0.00) | 25.0 (0.00) |
|       | 1.0 *(highest granularity)* | 54.1 (4.99) | **19.9** (1.34) | 25.2 (0.78) |

Table 7: Combined ablation studies for TCGA-UCEC. Sections ($\xi$: correlation threshold, $\tau$: group similarity. Bold marks the best in each category.

## Appendix I. Distribution of important features

We analyzed the distribution of the most important features, according to the attributions scores of Integrated Gradients. This highlighted the great value of nuclear features, as they are very prominent among the top-5 most attributed features.

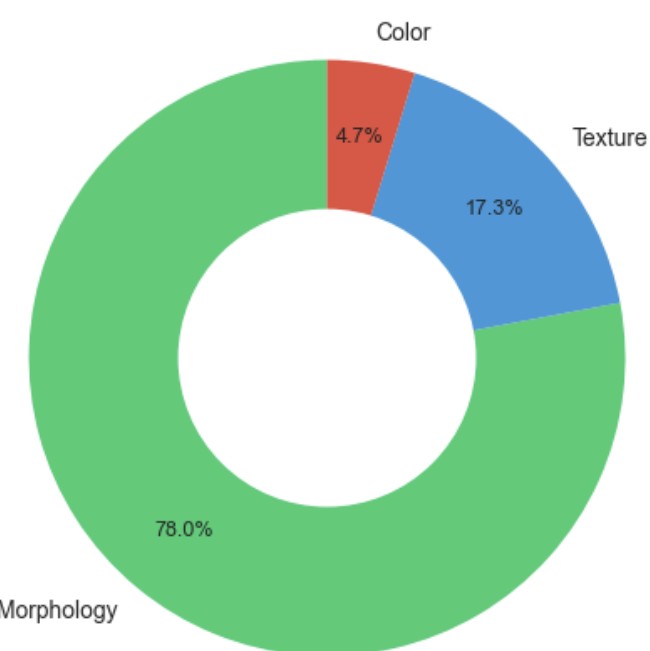

Figure 6: Distribution of features present in the top-5 most attributed features for TCGA-BRCA stage prediction.

## Appendix J. Full feature overview

Below is a complete list of all features and whether they are included for different correlation thresholds $\tau$.

| Feature Name | Corr $< 0.95$ | Corr $< 0.99$ | Feature Name | Corr $< 0.95$ | Corr $< 0.99$ |
|---|:---:|:---:|---|:---:|:---:|
| mean_r | ✓ | ✓ | median_r | ✓ | ✓ |
| mean_g | | ✓ | median_g | ✓ | ✓ |
| mean_b | | ✓ | median_b | ✓ | ✓ |
| mean_h | | ✓ | ratio_bright | ✓ | ✓ |
| mean_s | | ✓ | ratio_dark | ✓ | ✓ |
| mean_v | | ✓ | 10_dark | ✓ | ✓ |
| mean_l | | ✓ | 10_bright | ✓ | ✓ |
| mean_a | ✓ | ✓ | mean | ✓ | ✓ |
| mean_(la)b | ✓ | ✓ | size | ✓ | ✓ |

Table 8: List of region statistics and whether they are included for different correlation thresholds $\tau$.

| Feature Name | $< 0.95$ | $< 0.99$ | Feature Name | $< 0.95$ | $< 0.99$ |
|---|:---:|:---:|---|:---:|:---:|
| **original_firstorder** | | | **original_glrlm** | | |
| ...10Percentile | | ✓ | ...GrayLevelNonUniformity | ✓ | ✓ |

| Feature | | |
|---|---|---|
| ...90Percentile | ✓ | ✓ |
| ...Energy | ✓ | ✓ |
| ...Entropy | ✓ | ✓ |
| ...InterquartileRange | ✓ | ✓ |
| ...Kurtosis | ✓ | ✓ |
| ...Maximum | ✓ | ✓ |
| ...MeanAbsoluteDeviation | | ✓ |
| ...Mean | | |
| ...Median | | ✓ |
| ...Minimum | ✓ | ✓ |
| ...Range | | ✓ |
| ...RobustMeanAbsoluteDeviation | | ✓ |
| ...RootMeanSquared | | ✓ |
| ...Skewness | ✓ | ✓ |
| ...TotalEnergy | | |
| ...Uniformity | | ✓ |
| ...Variance | ✓ | ✓ |
| **original_glcm** | | |
| ...Autocorrelation | | ✓ |
| ...ClusterProminence | ✓ | ✓ |
| ...ClusterShade | ✓ | ✓ |
| ...ClusterTendency | ✓ | ✓ |
| ...Contrast | ✓ | ✓ |
| ...Correlation | ✓ | ✓ |
| ...DifferenceAverage | ✓ | ✓ |
| ...DifferenceEntropy | ✓ | ✓ |
| ...DifferenceVariance | ✓ | ✓ |
| ...Id | ✓ | ✓ |
| ...Idm | | |
| ...Idmn | ✓ | ✓ |
| ...Idn | ✓ | ✓ |
| ...Imc1 | ✓ | ✓ |
| ...Imc2 | ✓ | ✓ |
| ...InverseVariance | ✓ | ✓ |
| ...JointAverage | | |
| ...JointEnergy | | ✓ |
| ...JointEntropy | | ✓ |
| ...MCC | ✓ | ✓ |
| ...MaximumProbability | ✓ | ✓ |
| ...SumAverage | ✓ | ✓ |
| ...SumEntropy | | ✓ |

| Feature | | |
|---|---|---|
| ...GrayLevelNonUniformityNormalized | ✓ | ✓ |
| ...GrayLevelVariance | | ✓ |
| ...HighGrayLevelRunEmphasis | | ✓ |
| ...LongRunEmphasis | ✓ | ✓ |
| ...LongRunHighGrayLevelEmphasis | ✓ | ✓ |
| ...LongRunLowGrayLevelEmphasis | ✓ | ✓ |
| ...LowGrayLevelRunEmphasis | | ✓ |
| ...RunEntropy | ✓ | ✓ |
| ...RunLengthNonUniformity | ✓ | ✓ |
| ...RunLengthNonUniformityNormalized | ✓ | ✓ |
| ...RunPercentage | | ✓ |
| ...RunVariance | ✓ | ✓ |
| ...ShortRunEmphasis | ✓ | ✓ |
| ...ShortRunHighGrayLevelEmphasis | ✓ | ✓ |
| ...ShortRunLowGrayLevelEmphasis | ✓ | ✓ |
| **original_glszm** | | |
| ...GrayLevelNonUniformity | ✓ | ✓ |
| ...GrayLevelNonUniformityNormalized | ✓ | ✓ |
| ...GrayLevelVariance | ✓ | ✓ |
| ...HighGrayLevelZoneEmphasis | ✓ | ✓ |
| ...LargeAreaEmphasis | ✓ | ✓ |
| ...LargeAreaHighGrayLevelEmphasis | ✓ | ✓ |
| ...LargeAreaLowGrayLevelEmphasis | ✓ | ✓ |
| ...LowGrayLevelZoneEmphasis | ✓ | ✓ |
| ...SizeZoneNonUniformity | ✓ | ✓ |
| ...SizeZoneNonUniformityNormalized | ✓ | ✓ |
| ...SmallAreaEmphasis | ✓ | ✓ |
| ...SmallAreaHighGrayLevelEmphasis | ✓ | ✓ |
| ...SmallAreaLowGrayLevelEmphasis | ✓ | ✓ |
| ...ZoneEntropy | ✓ | ✓ |
| ...ZonePercentage | ✓ | ✓ |
| ...ZoneVariance | ✓ | ✓ |
| **original_ngtdm** | | |
| ...Busyness | ✓ | ✓ |
| ...Coarseness | ✓ | ✓ |
| ...Complexity | ✓ | ✓ |
| ...Contrast | ✓ | ✓ |
| ...Strength | ✓ | ✓ |

| | | | | | | |
|---|---|---|---|---|---|---|
| ...SumSquares | | | | | | |
| **original_gldm** | | | | | | |
| ...DependenceEntropy | ✓ | ✓ | | | | |
| ...DependenceNonUniformity | ✓ | ✓ | | | | |
| ...DependenceNonUniformity Normalized | | ✓ | | | | |
| ...DependenceVariance | ✓ | ✓ | | | | |
| ...GrayLevelNonUniformity | ✓ | ✓ | | | | |
| ...GrayLevelVariance | | | | | | |
| ...HighGrayLevelEmphasis | | | | | | |
| ...LargeDependenceEmphasis | | ✓ | | | | |
| ...LargeDependenceHighGray LevelEmphasis | ✓ | ✓ | | | | |
| ...LargeDependenceLowGray LevelEmphasis | ✓ | ✓ | | | | |
| ...LowGrayLevelEmphasis | ✓ | ✓ | | | | |
| ...SmallDependenceEmphasis | ✓ | ✓ | | | | |
| ...SmallDependenceHighGray LevelEmphasis | ✓ | ✓ | | | | |
| ...SmallDependenceLowGray LevelEmphasis | ✓ | ✓ | | | | |

Table 9: List of radiomics features and whether they are included for different correlation thresholds $\tau$.

| Feature Name | < 0.95 | < 0.99 | Feature Name | < 0.95 | < 0.99 |
|---|---|---|---|---|---|
| **all** | | | **inflam** | | |
| ...count | ✓ | ✓ | ...count | ✓ | ✓ |
| ...mean_area | ✓ | ✓ | ...mean_area | ✓ | ✓ |
| ...std_area | | ✓ | ...std_area | ✓ | ✓ |
| ...5p_area | ✓ | ✓ | ...5p_area | | |
| ...25p_area | ✓ | ✓ | ...25p_area | | ✓ |
| ...50p_area | | ✓ | ...50p_area | | |
| ...75p_area | | ✓ | ...75p_area | | |
| ...95p_area | | ✓ | ...95p_area | | |
| ...min_area | ✓ | ✓ | ...min_area | ✓ | ✓ |
| ...max_area | ✓ | ✓ | ...max_area | | ✓ |
| ...density | ✓ | ✓ | ...density | ✓ | ✓ |
| **nolabe** | | | **connec** | | |
| ...count | ✓ | ✓ | ...count | ✓ | ✓ |
| ...mean_area | ✓ | ✓ | ...mean_area | ✓ | ✓ |
| ...std_area | ✓ | ✓ | ...std_area | ✓ | ✓ |
| ...5p_area | | | ...5p_area | | |
| ...25p_area | | | ...25p_area | ✓ | ✓ |
| ...50p_area | | | ...50p_area | | ✓ |
| ...75p_area | | | ...75p_area | | ✓ |
| ...95p_area | | | ...95p_area | | ✓ |
| ...min_area | | ✓ | ...min_area | ✓ | ✓ |
| ...max_area | | ✓ | ...max_area | ✓ | ✓ |
| ...density | ✓ | ✓ | ...density | ✓ | ✓ |
| **neopla** | | | **necros** | | |
| ...count | ✓ | ✓ | ...count | ✓ | ✓ |
| ...mean_area | ✓ | ✓ | ...mean_area | ✓ | ✓ |
| ...std_area | ✓ | ✓ | ...std_area | ✓ | ✓ |
| ...5p_area | | ✓ | ...5p_area | | |
| ...25p_area | | ✓ | ...25p_area | | ✓ |
| ...50p_area | | | ...50p_area | | |
| ...75p_area | | | ...75p_area | | |
| ...95p_area | | ✓ | ...95p_area | | |
| ...min_area | ✓ | ✓ | ...min_area | ✓ | ✓ |
| ...max_area | ✓ | ✓ | ...max_area | ✓ | ✓ |
| ...density | ✓ | ✓ | ...density | ✓ | ✓ |
| | | | **no-neo** | | |
| | | | ...count | ✓ | ✓ |
| | | | ...mean_area | ✓ | ✓ |
| | | | ...std_area | ✓ | ✓ |
| | | | ...5p_area | | |
| | | | ...25p_area | | ✓ |
| | | | ...50p_area | | |
| | | | ...75p_area | | |
| | | | ...95p_area | | |
| | | | ...min_area | ✓ | ✓ |
| | | | ...max_area | | ✓ |
| | | | ...density | ✓ | ✓ |

Table 10: List of nuclear characteristics and whether they are included for different correlation thresholds $\tau$.

