# OpenReview forum: "From Pixels to Histopathology: A Graph-Based Framework for Interpretable Whole Slide Image Analysis"
_MIDL.io/2026/Conference — MIDL 2026 Poster_

### Official Review · Reviewer_UBxu · 2026-01-01

**Confidence:** 5
**Preliminary Rating:** 3

**Summary:**

This paper presents a graph-based framework for Whole Slide Image analysis that utilizes superpixels to ensure nodes respect natural biological boundaries. The authors introduce an adaptive coarsening strategy to merge homogeneous tissue regions while retaining high granularity in diagnostically critical areas. Nodes are enriched with interpretable clinical features covering nuclear, texture, and morphological domains, enabling diagnosis via a graph attention network. Experiments on cancer staging and survival prediction demonstrate superior performance over standard baselines.

**Strengths:**

- A major strength of this paper is the transition from rigid rectangular grids to a graph structure based on superpixels. This approach better respects natural tissue boundaries and preserves the structural context essential for pathological assessment.
- The proposed adaptive coarsening strategy is technically sound and innovative. It effectively balances computational efficiency with diagnostic granularity by merging homogeneous areas while retaining high resolution in complex, heterogeneous regions.
- Unlike many black-box deep learning models, this framework is interpretable by design. By utilizing domain-inspired features such as nuclear statistics and texture descriptors, the model provides explanations that are grounded in clinical knowledge and intuitive for pathologists.

**Weaknesses:**

- The graph construction and coarsening process rely on predefined similarity thresholds. The authors do not provide an extensive analysis of how sensitive the final classification performance is to these thresholds across different tissue types or staining variations, which could impact the robustness of the method in diverse clinical settings.
- While the results on the selected cohorts are promising, the evaluation is primarily restricted to two types of carcinoma. Expanding the validation to include a wider variety of organs and non-epithelial tumors would further substantiate the generalizability of the proposed graph representation.
- The authors' findings indicate that on the smaller dataset, the performance gap between the proposed method and existing graph-based baselines is relatively narrow, particularly regarding certain classification metrics where it does not achieve a clear lead.

**Detailed Comments:**

- The parameters for graph coarsening and feature pruning appear to be determined via hyperparameter sweeps. A discussion on whether these values remain stable across different organ types or if they require task-specific tuning would offer better guidance for researchers applying the framework to new datasets.
- In the qualitative analysis sections, specific "Important Features" are highlighted. It would be insightful to discuss whether these attributions are consistent across a larger subset of the cohort or if they vary significantly between individual samples of the same cancer stage.

**Justification Of The Preliminary Rating:**

The paper presents a promising approach by replacing rigid grids with structure-adaptive graphs that respect biological boundaries. This transition provides a strong structural prior, leading to impressive data efficiency and intrinsic interpretability compared to standard methods. However, there are concerns regarding the heavy computational overhead of the multi-stage preprocessing and the model's sensitivity to predefined thresholds. Furthermore, while the feature attributions are intuitive, their clinical utility lacks formal validation by pathologists.

**Questions To Address In The Rebuttal:**

- While the feature attributions are shown to be clinically relevant, have the authors conducted any qualitative assessments with board-certified pathologists to confirm that the regions and features highlighted by the model truly align with professional diagnostic criteria?
- The current framework primarily focuses on node features. Did the authors experiment with incorporating edge features, such as the spatial distance between region centroids or the length of shared boundaries? Would adding such topological information further enhance the model's structural awareness?

---

> ### Author Response · Authors · 2026-01-24
>
> We thank the reviewer for their insightful comments regarding the robustness of our graph construction and the consistency of our explanations. We have addressed these points with new sensitivity analyses.
>
>
> **R UBxu**: *The graph construction and coarsening process rely on predefined similarity thresholds. The authors do not provide an extensive analysis of how sensitive the final classification performance is to these thresholds across different tissue types or staining variations, which could impact the robustness of the method in diverse clinical settings.*
>
> **TL;DR Answer**: We added an ablation on TCGA-UCEC (Appendix H), confirming robustness. Since this step occurs after preprocessing, tuning on new datasets is computationally inexpensive.
>
> We have expanded our sensitivity analysis on the merging threshold to include the TCGA-UCEC dataset (see Appendix H). The results indicate that our method remains robust across datasets, and notably, slightly larger regions performed well on UCEC. Furthermore, because graph construction and coarsening occur after the computationally heavy feature extraction, running these ablations is very fast. Therefore, while the method is robust, we recommend a quick sweep of this threshold when applying the framework to new datasets to maximize performance (Section 4.4). We have also noted validation on staining variations as a direction for future work in Section 5.
>
> **R UBxu**: *The parameters for graph coarsening and feature pruning appear to be determined via hyperparameter sweeps. A discussion on whether these values remain stable across different organ types or if they require task-specific tuning would offer better guidance for researchers applying the framework to new datasets.*
>
> **TL;DR Answer**: We observed similar optimal value ranges for both BRCA and UCEC. However, given the low computational cost, we recommend dataset-specific tuning.
>
> Our experiments on both TCGA-BRCA and TCGA-UCEC suggest that the optimal ranges for coarsening and pruning parameters are relatively stable across different organ types. We did not require task-specific tuning to achieve strong performance on either dataset. However, consistent with our response above, because the computational overhead for these specific parameters is negligible compared to the initial feature extraction, we recommend performing a rapid dataset-specific optimization to ensure the best possible results for new cohorts.
>
> **R UBxu**: *In the qualitative analysis sections, specific "Important Features" are highlighted. It would be insightful to discuss whether these attributions are consistent across a larger subset of the cohort or if they vary significantly between individual samples of the same cancer stage.*
>
> **TL;DR Answer**: We added a distribution overview in Appendix I. Nuclear features consistently dominate the attributions, though specific values vary by sample, enabling cohort-level insights.
>
> We have included an overview of the distribution of the most attributed features for stage prediction on TCGA-BRCA in Appendix I. This analysis reveals that nuclear features consistently dominate the importance rankings across the cohort, although the specific feature values vary between individual samples as biologically expected. This distribution of features itself could be valuable for future cohort-level analysis.
>
> **R UBxu**: *While the feature attributions are shown to be clinically relevant, have the authors conducted any qualitative assessments with board-certified pathologists to confirm that the regions and features highlighted by the model truly align with professional diagnostic criteria?*
>
> **TL;DR Answer**: Expert validation confirmed that the model's logic aligns with diagnostic criteria, while noting that clinical staging is often multi-modal (e.g., including lymph nodes), unlike the WSI-only model.
>
> We consulted a pathologist to evaluate the explanations our method provided. We received positive feedback confirming that both the selected regions and the highlighted features (e.g., nuclear atypia) align well with professional diagnostic criteria. However, the expert importantly noted that a direct comparison to clinical workflow has limitations: in clinical practice, stage grading is rarely done based on a single WSI alone but often incorporates other modalities (e.g., lymph node examinations in breast cancer). We have updated Section 4.5 to reflect this valuable nuance, highlighting that our model correctly identifies the WSI-based morphological evidence for staging.

---

> ### Author Response · Authors · 2026-01-24
>
> **R UBxu**: *The current framework primarily focuses on node features. Did the authors experiment with incorporating edge features, such as the spatial distance between region centroids or the length of shared boundaries? Would adding such topological information further enhance the model's structural awareness?*
>
> **TL;DR Answer**: We experimented with edge features (e.g., boundary length) and heterogeneous graphs, but they increased complexity without improving performance.
>
> We conducted experiments incorporating edge attributes, such as the length of shared boundaries, and exploring heterogeneous graph structures where edge types were determined by the dominant nuclei classes of connected regions. We found that these additions increased the computational complexity of the model without yielding any statistically significant performance gains.

---

### Official Review · Reviewer_wQyq · 2026-01-06

**Confidence:** 5
**Preliminary Rating:** 4
**Final Rating:** 5

**Summary:**

The authors proposed a graph-based framework for analysing gigapixel whole-slide images. This framework provides a pipeline that builds a too-fine-grained graph from a whole-slide image, extracts a coarse graph from the fine-grained graph, embeds feature vectors into the coarse graph’s nodes, and applies a graph attention network to classify embedded features on the extracted coarse graph. Using integrated gradients (one of the common visual explanation methods), the authors aim to interpret the model's predictions by highlighting an important region and the statistical characteristics of embedded feature vectors on the coarse graph.

**Strengths:**

- The graph-based approach mitigates the issue of destroying biological structures in previous patch-based processing.
- The proposed pipeline's coarsening strategy extracts a computable-size graph, which approximately represents the structure of an input whole slide image, from a gigapixel whole slide image for the graph attention network's computation towards further downstream tasks.
- The proposed pipeline achieves superior and competitive results compared to previous works for cancer stage classification and patient survival prediction.

**Weaknesses:**

- Mathematically incorrect notations and bumpy definitions exist.
- Whether the extracted compact graph truly expresses the biological structure or not is unclear.
- Unclear details of feature vectors.
- Unclear interpretability of the proposed pipeline.

**Detailed Comments:**

The proposed framework focuses on processing of gigapixel whole-slide images without cropping them into small patches. Using superpixel segmentation and a coarsening procedure based on local geometric appearances, the framework extracts biological structure as a graph representation, where homogeneous and heterogeneous regions are represented by sparse and dense nodes, respectively.  This graph representation mitigates computational scalability issues and presents biological structure within a whole-slide image. This is an interesting machine-learning-based analytical method for digital pathology, even though the accuracy of stage classification and survival prediction appears insufficient for real-world application (based on results from recent publications).

The interpretability of the proposed framework should require more detailed analysis and discussion, as current definitions and explanations of the features and their analyses are unclear. If the authors think interpretability is important, this point is non-negligible. For stage classification, populations (distributions) of nuclei subtypes should be an essential feature, as reported in many previous works. However, the interpretation of the model's processing via integrated gradients is unclear. Since the proposed framework also includes feature pruning, the full pipeline becomes complex; therefore, the analysis procedure should be described in more detail.

At p. 5, "x_i = x^{\mathrm{nuc}}_i \|  x^{\mathrm{tex}}_i   \|  x^{\mathrm{morph}}_i" does not make sense, since \| \| is undefined operator here. If the authors want to concatenate three  column vectors, the following is one way with generic mathematical notation: "x_i = [ x^{\mathrm{nuc}\top}_i,  x^{\mathrm{tex}\top}_i, x^{\mathrm{morph}\top}_i ]^{\top} \in \mathbb{R}^{191}" Even though the authors might think it look not cool notation, this standard notation is much better than the current meaningless description.

Training setting for a model in which the loss function and hyperparameters are unclear. Please provide clear definitions.

**Justification Of Final Rating:**

During the rebuttal and discussion periods, the authors answered all of my questions. Furthermore, they revised their manuscript in response to my comments on the weaknesses. This submission is now ready, with its strength and solid manuscript, for the MIDL presentation this year.

**Justification Of The Preliminary Rating:**

The main idea of coarsening a too fine-grained initial graph looks like an interesting and useful approach for gigapixel whole-slide images in the future. Even though there are some unclear points, the manuscript structure is solid, and the concept is clear. The experimental setting and results, including comparative evaluations and an ablation study for two datasets, are also convincing.

**Questions To Address In The Rebuttal:**

- For node feature extraction, does each node have the same local region(size)? In Fig. 1, superpixel regions have different patch sizes. The details of feature extraction are unclear.
- In Fig. 1, how does the framework produce the corresponding segmentation map(above 6) Full WSI Graph)?
- In Fig. 2, only small regions are marked up and explained, but are they the only important regions for real diagnosis?
- In this work, did the authors work with pathologists, and did the authors discuss the results in this manuscript with pathologists?
- Even though two of three features  are mathematically defined statistics and the residual one offers predicted nuclei information, how to interpret and make the final interpretation, such as the results shown in Fig. 2? The current approach has the freedom to select an interpretation way from the suggested information in the pipeline. In other words, the proposed pipeline does not offer a unique interpretation.

---

> ### Author Response · Authors · 2026-01-24
>
> We thank the reviewer for their detailed assessment and for raising important questions regarding the interpretability and clinical validity of our method. We have included additional data and expert validation to address these points.
>
> **R wQyq**: *For stage classification, populations (distributions) of nuclei subtypes should be an essential feature, as reported in many previous works.*
>
> **TL;DR Answer**: Nuclei subtype counts are explicitly included in our feature set and are consistently identified as high-importance drivers by the model.
>
> We fully agree. As clarified in the revised Section 3.3, our statistics on nuclei include both the total count and the count per subtype, so the model has this population information. Our results confirm the importance of these features, as they are frequently assigned high attribution scores by the model (see Section 4.5 and Appendix I).
>
> **R wQyq**: *Since the proposed framework also includes feature pruning, the full pipeline becomes complex*
>
> **TL;DR Answer**: Feature pruning is a one-time global operation on the training set, not a complex per-sample inference step.
>
> We wish to clarify that feature pruning is a one-time operation, not a sample-level preprocessing step. We computed the correlation matrix once on the training set to identify and remove redundant features from the pipeline entirely. We have updated Section 3.3 to make this distinction clear.
>
>
> **R wQyq**: *Incorrect mathematical notation for concatenating three column vectors*
>
> **TL;DR Answer**: Thanks, corrected.
>
> We thank the reviewer for spotting this notation error. We have corrected the concatenation formulation in Section 3.3.
>
>
> **R wQyq**: *Training setting for a model in which the loss function and hyperparameters are unclear.*
>
> **TL;DR Answer**: Added Appendix F with all hyperparameters and clarified loss functions in Section 4.1.
>
> We have added Appendix F, which lists all hyperparameters and search spaces. Additionally, we revised Section 4.1 to explicitly state the loss functions used (Cross-Entropy for staging, NLL-Survival loss for survival prediction).
>
>
> **R wQyq**: *For node feature extraction, does each node have the same local region(size)? In Fig. 1, superpixel regions have different patch sizes.*
>
> **TL;DR Answer**: Regions vary within a consistent range (see new Appendix G), and region size is explicitly included as a node feature.
>
> The superpixels generated by SLIC vary in size as they adapt to local tissue boundaries, though they remain within a consistent magnitude. We have added Appendix G (Table 6) to show the distribution of region sizes. Furthermore, "region size" is included as an explicit feature in the node vector, allowing the model to account for these variations during prediction.
>
>
> **R wQyq**: *In Fig. 1, how does the framework produce the corresponding segmentation map(above 6) Full WSI Graph)?*
>
> **TL;DR Answer**: The "corresponding segmentation map" shows the initial superpixel segmentation after the merging process.
>
> The "corresponding segmentation" is based on the initial segmentation and the merging of similar, adjacent nodes. It is not explicitly materialized during training or inference, but is implicitly present and can be reconstructed from the saved initial segmentation and merging operations.
>
>
> **R wQyq**: *In Fig. 2, only small regions are marked up and explained, but are they the only important regions for real diagnosis?*
>
> **TL;DR Answer**: Highlighted regions are the top predictive drivers. Pathologist validation confirms they align with diagnostic areas, though clinical review is often broader.
>
> The highlighted regions in Figure 2 represent the specific areas that most strongly influenced the model's prediction. We validated these with a pathologist (see Section 4.5), who confirmed that these regions generally align with diagnostically relevant areas (e.g., tumor boundaries, inflammatory infiltrates).
>
>
> **R wQyq**: *In this work, did the authors work with pathologists, and did the authors discuss the results in this manuscript with pathologists?*
>
> **TL;DR Answer**: Yes, we collaborated with a pathologist who validated the clinical relevance of our regions, features, and explanations (see extended Section 4.5).
>
> We actively collaborated with a pathologist to validate our method. We have added details to Section 4.5 that outline their feedback. The expert positively evaluated the tissue-aligned segmentation and the interpretability of the feature set, noting that the model's focus on specific morphological signals (like nuclear density or artifacts) allows for a level of verification that is impossible with black-box methods.

---

> > ### Comment · Reviewer_wQyq · 2026-01-29
> >
> > Thank you for your feedback; the authors fully addressed my comments, except for the following point.
> >
> > >The "corresponding segmentation map" shows the initial superpixel segmentation after the merging process.
> >
> > Please add a description about this point to the manuscript if it is possible.

---

> > > ### Author Response · Authors · 2026-01-29
> > >
> > > Thank you for responding before the extended PDF update deadline. Thus, we were able to include both suggestions: we have added a concise description of Integrated Gradients and its usage to the Method section (see new Section 3.5) and updated the caption for Figure 1 to be clearer and more complete. We hope these revisions fully clarify the points raised and address your concerns regarding the method's presentation.

---

> > > > ### Comment · Reviewer_wQyq · 2026-01-29
> > > >
> > > > Thank you for prompt feedback. That's fully satisfactory.
> > > > Many thanks for discussion.

---

> ### Author Response · Authors · 2026-01-24
>
> **R wQyq**: *Even though two of three features are mathematically defined statistics and the residual one offers predicted nuclei information, how to interpret and make the final interpretation, such as the results shown in Fig. 2? The current approach has the freedom to select an interpretation way from the suggested information in the pipeline.In other words, the proposed pipeline does not offer a unique interpretation.*
>
> **TL;DR Answer**: The attribution scores are mathematically unique to the model's weights and input. We show the features and regions with the highest scores.
>
> We would like to clarify that the interpretation is not heuristically selected. The features highlighted in Figure 2 are derived using Integrated Gradients, which provides a mathematically unique attribution score for every feature given the trained model weights and specific input. Therefore, the underlying ranking of importance is deterministic and unique to the model's decision process.

---

> > ### Comment · Reviewer_wQyq · 2026-01-29
> >
> > Thank you for your feedback. Based on your reply, I rechecked and understood what you mentioned, integrated gradients. To clarify this point in the manuscript, please provide a concise explanation of integrated gradient usage, including attribution score-based processing, in the Methodology section (not the experiment section) if it is possible. Necessary information at hand should require appropriate sectioning.

---

### Official Review · Reviewer_xuVJ · 2026-01-10

**Confidence:** 5
**Preliminary Rating:** 4
**Final Rating:** 4

**Summary:**

This paper describes a whole slide image classification method with multiple innovations including the use of superpixels, graph coarsening, a graph attention network and interpretable features. A key advantage of this approach is the use of natural structure borders rather than an arbitrary grid. The method is evaluated on a grading and survival prediction tasks on two TCGA datasets and shows promising results relative to a few baseline methods.

**Strengths:**

Aside from a few missing details in the method (see detailed comments), this paper is very clearly presented and well organized. The method is well motivated and addresses multiple issues with prevailing MIL approaches, such as lack of interpretability and use of an arbitrary grid. The results seems promising relative to various baslines.

**Weaknesses:**

Assuming that the patch features used for the baselines are different to "interpretable" feature vector used in the proposed method, this makes it difficult to discern whether the features vectors or the architectures (or both) give rise the improved performance. This is an important missing ablation study that makes it rather difficult to interpret the significance of the results.

Though it is not discussed in the paper, I would imagine that running inferences for this method is rather slow and complex, involving multiple steps including running HoVerNet, feature calculations, the iterative graph coarsening, etc.

The use of integrated gradients for feature attribution is limited and will fail to capture significant relationships between different features and/or regions (nodes) that give rise to the prediction. I think this should probably be discussed in the paper

**Detailed Comments:**

There are a few parts of the method that are missing detail:
- It is not clear how a the ResNet-18 can generate an embedding for an irregularly-shaped region, which would require a rectangular input.
- During the coarsening process, how is the embedding of the new, merged node calculated from the embeddings of the predecessor nodes?
- How is the aggregation done in section 3.4? The text makes it sound like the node embeddings are concatenated, but this would give a different size vector depending on the number of nodes in the graph, which would not feed into an MLP.
- The performance of the baseline methods will depend substantially on the choice of patch embeddings. However this is not explicitly stated for ABMIL, DM-GNN etc.

**Justification Of Final Rating:**

The authors have clarified many points, however I am confused by their experiment comparing to "CLAM" features, since the CLAM model just uses ImageNet-trained ResNet features, and beating these is not especially interesting in my opinion. I have decided to leave my overall rating unchanged as a result

**Justification Of The Preliminary Rating:**

There are many strong aspects to this paper, however I am a little hesitant because it is impossible to know given the current experiments whether it is the model architecture or feature set (or both) that give rise to the improvement over baselines.

**Questions To Address In The Rebuttal:**

The most important question to answer is which patch features were used for the baseline methods.

See other questions in detailed comments.

---

> ### Author Response · Authors · 2026-01-24
>
> We thank the reviewer for their constructive feedback. We have addressed the concerns regarding the ablation of features versus architecture and clarified our design choices regarding graph construction and interoperability.
>
> **R xuVJ**: *Assuming that the patch features used for the baselines are different to "interpretable" feature vector used in the proposed method, this makes it difficult to discern whether the features vectors or the architectures (or both) give rise the improved performance. This is an important missing ablation study that makes it rather difficult to interpret the significance of the results.*
>
> **TL;DR Answer**: We added this crucial ablation (Table 2). Results show that our performance is driven by the synergy of interpretable features and graph architecture.
>
> We agree that disentangling the contributions of the architecture from those of the feature set is crucial. We have added this ablation study to the manuscript (4.4 and Table 2). We evaluated all combinations of learned embeddings (CLAM [1]) vs interpretable features, and MIL architecture vs graph-based approach. The results demonstrate that while interpretable features perform comparably to learned embeddings in a standard MIL setting, their predictive power is significantly boosted when contextualized within a graph architecture. Conversely, learned embeddings (CLAM) do not benefit significantly from the graph structure in our experiments. This confirms that our method's performance is driven by the combination of our feature set and the graph architecture.
>
> | Dataset | Model | Input Features | AUC ↑ | F1ₘ ↑ | Bal. Acc ↑ | C-Index ↑ |
> |--------|-------|----------------|-------|-------|------------|-----------|
> | BRCA | ABMIL† | CLAM | _61.8*_ (5.64) | _20.2*_ (2.13) | 26.0 (1.11) | _61.7_ (1.81) |
> | BRCA | ABMIL | Interpretable | 58.9* (5.28) | 16.8* (4.49) | _26.7_ (2.95) | 51.0* (2.30) |
> | BRCA | Graph | CLAM | 57.9* (5.65) | 19.1* (0.37) | 25.1 (0.12) | 51.4* (1.14) |
> | BRCA | **Graph†** | **Interpretable (Ours)** | **67.2** (3.08) | **28.0** (8.24) | **27.0** (2.54) | **62.9** (3.67) |
> | UCEC | ABMIL† | CLAM | 49.0* (2.61) | 18.8 (0.00) | 25.0 (0.00) | 58.3 (5.31) |
> | UCEC | ABMIL | Interpretable | _52.1_ (2.74) | 17.6 (3.59) | 25.1 (1.63) | 51.3 (2.87) |
> | UCEC | Graph | CLAM | 52.0 (3.49) | **20.2** (1.73) | _25.4_ (0.92) | _59.6_ (4.18) |
> | UCEC | **Graph†** | **Interpretable (Ours)** | **56.4** (4.17) | _19.5_ (1.19) | **26.5** (2.37) | **60.0** (0.97) |
>
>
> **R xuVJ**: *Though it is not discussed in the paper, I would imagine that running inferences for this method is rather slow and complex.*
>
> **TL;DR Answer**: Preprocessing is a one-time, parallelizable cost. Subsequent graph inference is nearly instantaneous and can be integrated seamlessly into clinical digitization workflows.
>
> While the preprocessing pipeline involves superpixel segmentation and feature extraction (HoVerNet), these are one-time, parallelizable costs. Once the slide is converted into our compressed graph representation, the actual inference is lightweight and nearly instantaneous. In a clinical setting, slides are typically digitized and preprocessed before review. Our pipeline can be integrated into this phase, ensuring no additional latency during the pathologist's actual workflow.
>
>
> **R xuVJ**: *The use of integrated gradients for feature attribution is limited and will fail to capture significant relationships between different features and/or regions.*
>
> **TL;DR Answer**: While IG is additive, it reflects node contributions given the structure-aware embeddings learned by the GAT layers. Pathologist validation confirms its utility.
>
> We acknowledge that Integrated Gradients (IG) is an additive method and does not explicitly model feature interactions. However, we argue it remains a highly valuable tool for our analysis:
> - IG satisfies the sensitivity axiom, ensuring that any feature contributing to the prediction is attributed.
> - Since our model uses GAT layers, the node embeddings at the readout stage already contain aggregated contextual information from neighbors. Therefore, the IG attribution score for a node reflects its contribution given the learned structural interactions.
> - As detailed in the extended section 4.5, a pathologist confirmed that the regions and features highlighted by IG align well with diagnostic criteria, validating its utility as a first-order explanation tool.
>
> [1] Lu, M.Y., Williamson, D.F.K., Chen, T.Y. et al. Data-efficient and weakly supervised computational pathology on whole-slide images. Nat Biomed Eng 5, 555–570 (2021). https://doi.org/10.1038/s41551-020-00682-w

---

> > ### Comment · Reviewer_xuVJ · 2026-02-01
> >
> > I like the spirit of this response, but it leaves me a little bewildered...
> >
> > "CLAM" features don't really exist. CLAM is a method to aggregate such features, rather than a feature extractor itself. The original paper, referenced in the comment above, uses a ResNet50 pretrained on ImageNet as a tile feature encoder. Thus there are no features specific to CLAM.
> >
> > So what are the feature vectors used in the above table? I suppose they are just ResNet-50 features, in which case it is not especially interesting that the interpretable features are comparable. There are many open-weight pathology-specific encoders available now (CONCH, UNI, CTransPath, CHIEF, ...) that outperform an ImageNet-pretrained ResNet

---

> > > ### Author Response · Authors · 2026-02-01
> > >
> > > Thanks for your response. We agree that our phrasing was imprecise. As you noted, the features in question are the standard ResNet50 (ImageNet) embeddings used as the default in the original CLAM paper.
> > >
> > > We will correct the terminology in the final camera-ready manuscript

---

> ### Author Response · Authors · 2026-01-24
>
> **R xuVJ**: *It is not clear how a the ResNet-18 can generate an embedding for an irregularly-shaped region, which would require a rectangular input.*
>
> **TL;DR Answer**: We extract bounding boxes and zero-mask the background.
>
> We have clarified this in Section 3.2. To process irregular regions with a ResNet-18, we extract the bounding box of that region and apply a zero-mask (black out) all pixels outside the superpixel boundary. This isolates the visual features of the specific region while satisfying the rectangular input requirement. A consulting pathologist confirmed that the resulting regions, when merged based on these embeddings, form biologically coherent structures.
>
> **R xuVJ**: *During the coarsening process, how is the embedding of the new, merged node calculated from the embeddings of the predecessor nodes?*
>
> **TL;DR Answer**: We keep the initial embeddings of each node to merge based on local similarity, preserving large homogeneous areas with slight gradients.
>
> We have clarified the coarsening logic in Section 3.2. Crucially, we do not update embeddings when nodes are merged, but instead continue to use every node's initial embedding to determine merging with it's directly neighboring regions. The decision to merge is based solely on the similarity of the original local embeddings of adjacent constituent nodes. This greedy agglomerative approach allows the algorithm to group large homogeneous areas (which may exhibit gradual staining gradients) based on local connectivity. If the embeddings were to be recomputed or averaged, two adjacent, similar regions might not be merged due to tissue features far away from the regions shared border.
>
> **R xuVJ**: *How is the aggregation done in section 3.4?*
>
> **TL;DR Answer**: We use standard mean pooling.
>
> We have revised Section 3.4 to explicitly state that we use a standard mean pooling aggregation, which empirically showed robust performance in our experiments.
>
> **R xuVJ**: *The performance of the baseline methods will depend substantially on the choice of patch embeddings. However this is not explicitly stated for ABMIL, DM-GNN etc.*
>
> **TL;DR Answer**: We use the embeddings that are mentioned in the respective papers (DM-GNN: CLAM, GraphTransformer: custom pre-trained model), and for DeepSets and ABMIL we use CLAM embeddings according to relevant literature.
>
> We have revised the manuscript to explicitly clarify the patch embeddings used (Section 4.1):
> - GraphTransformer: Uses a custom pre-trained model that was made publicly available by the authors.
> - DM-GNN: We follow the paper and use CLAM embeddings to reproduce it.
> - DeepSets and ABMIL: Both are embedding agnostic, so we follow relevant literature (e.g., DM-GNN, GraphTransformer) and use CLAM embeddings.
>
> This embedding choice, especially when combined with the new ablation on CLAM embeddings in a graph architecture (Section 4.4), enables a fair comparison.

---

### Author Rebuttal · Authors · 2026-01-24

**Rebuttal:**

Dear Reviewers,

We would like to thank you all for your insightful and constructive feedback. We have carefully addressed each comment, provided detailed clarifications in the individual responses below, and revised the manuscript accordingly (changes are in red).

Summary of Changes:

- **Added Architecture vs. Features Ablation**: We included a new study (Section 4.4, Table 2) disentangling the contribution of our graph architecture from the interpretable feature set, demonstrating that their synergy drives performance.

- **Clarified Baselines**: We explicitly detailed the feature sets used for all baselines in Section 4.1 to ensure a fair and transparent comparison.


- **Detailed Methodology**: We clarified the graph coarsening logic (Section 3.2), pooling operations (Section 3.4), handling of irregular regions for feature extraction, and a concise description of the usage of Integrated gradients for explainability (Section 3.5).


- **Expanded Validation**: We added a qualitative expert review with a pathologist (Section 4.5) validating our interpretability claims and feature attributions.


- **New Appendices**: We added comprehensive appendices covering hyperparameters (Appendix F), region size statistics (Appendix G), additional ablations on UCEC (Appendix H), and an important feature distribution analysis (Appendix I).

- Minor clarifications.

We look forward to engaging further during the discussion phase to answer any remaining questions.

Many thanks and best wishes,

the authors

**Supporting Material:**

/attachment/57a673073665eed7dd088116c6db8f8e19e6a871.pdf

---

### Author Response · Authors · 2026-01-30

Dear Reviewers,

as we approach the end of the discussion period (Feb 1st), we wanted to gently check if you have had a chance to review our detailed responses. We put considerable effort into the rebuttal to ensure that all points raised were addressed thoroughly, and we improved the manuscript's quality in response to your feedback.

We would appreciate a brief comment indicating if our responses have adequately addressed your concerns, or if you require any final clarifications before the deadline.

Thank you again for your time and expertise.

Best,
the authors

---

### Meta-Review · Area_Chair_HJu2 · 2026-02-07

**Recommendation:** Accept (Oral)
**Confidence:** 5

**Metareview:**

This paper proposes a graph-based approach for the problem of gigapixel images analysis, such histopathology images. The typical patch-based methods are a computational fix to address memory limitations, but it is not grounded in any biological reality, quite the contrary.

The proposed method in stead builds an interpretable graph of the biological features. While performances do not seem to be state-of-the-art yet, it is a very welcome and needed departure from other grid + MIL based approaches.

In my opinion this paper strongly deserves an oral spot at the conference. I also see that the authors have already shared anonymously their code ; I urge them to make their method as reproducible and _reusable_ as possible, if not already the case.

---

### Decision · Program_Chairs · 2026-02-13

Accept (Poster)